# Valproic Acid as a Histone Deacetylase Inhibitor Induces *ABCB1* Overexpression and De Novo *ABCB5* Expression in HeLa Cells

**DOI:** 10.3390/cimb47090749

**Published:** 2025-09-11

**Authors:** Gabriela Rebeca Luna-Palencia, José Correa-Basurto, Ismael Vásquez-Moctezuma

**Affiliations:** 1Departamento de Biotecnología y Bioingeniería, Centro de Investigación y de Estudios Avanzados del Instituto Politécnico Nacional, Mexico City 07360, Mexico; galuna@cinvestav.mx; 2Laboratorio de Diseño y Desarrollo de Nuevos Fármacos e Innovación Biotecnológica, Escuela Superior de Medicina del IPN, Mexico City 11340, Mexico; jcorreab@ipn.mx; 3Laboratorio de Morfología, Escuela Superior de Medicina del IPN, Mexico City 11340, Mexico

**Keywords:** ABCB5, MDR1, valproic acid, epigenetic drugs, anticancer drugs

## Abstract

Histone deacetylase inhibitors (HDACis) induce the expression of multidrug resistance (MDR) pumps and can even display the MDR phenotype in cell lines. This is the first report to include the profiles of ATP-binding cassette (ABC) transporters in intrinsically expressed HeLa cells as well as those acquired due to a 5 mM valproic acid (VPA) treatment. Expression of ABC transporters related to the MDR phenotype was analyzed by RT-PCR in untreated HeLa cells and HeLa cells treated with 5 mM VPA. The ABCB5 protein was identified in HeLa cells by immunocytochemistry. HeLa cell treatment with 5 mM VPA increased *ABCB1* expression and triggered the de novo expression of *ABCB5* in mRNA and protein. Despite the expression of *ABCB5* and the overexpression of *ABCB1*, VPA reduced the growth rate by 20%, delayed doubling time by 25%, and decreased the number of living cells per well to 50% after 72 h. Pretreatment with VPA for 24 h followed by cotreatment with doxorubicin (DOX) sensitized HeLa cells to DOX. However, for the de novo expression of *ABCB5*, HeLa cells did not acquire the MDR phenotype from the 5 mM VPA treatment. The ABCB5 isoform induced by VPA treatment probably lacks MDR activity.

## 1. Introduction

It was discovered that daunomycin can be extruded by an active transport mechanism in resistant Ehrlich ascites tumor cells [1] and that the drug-resistant phenotype observed in colchicine-resistant Chinese hamster cells was due to a cell surface glycoprotein known as P-glycoprotein, revealing insights into the MDR world [2].

Failure in the use of diverse therapies for cancer, rheumatic arthritis, bacterial infections, and epilepsy has been attributed to MDR. In cancer, MDR is an intrinsic or acquired mechanism whereby cells show resistance to one drug, which is accompanied by resistance to other structurally different chemotherapeutic drugs. Intrinsic resistance is a pre-existing mechanism that decreases the initial efficacy of anticancer treatment and is independent of any prior exposure to cytotoxic drugs, whereas acquired resistance is induced by repeated drug administrations. The MDR phenotype is mainly characterized by the overexpression of ABC drug transporter proteins, which facilitate the efflux of cytotoxic agents from the cell via an ATP-dependent process, resulting in a lower intracellular accumulation of anticancer drugs. ABC transporters are not only expressed by cancer cells; they are also located in the cell membrane of many normal tissues to protect the cell from cytotoxic agents [3,4]. The human genome includes 48 ABC genes that encode for these transmembrane transporters, which are classified into seven families, from ABCA to ABCG, based on the sequence and organization of the nucleotide-binding domain (NBD) and one pseudogene known as *ABCC13* [5,6]. ABC transporters can be found as complete transporters, with two transmembrane domains (TMDs) and two nucleotide-binding domains (NBDs) for ATP binding and hydrolysis, or as half transporters, which have to homo- or heterodimerize to be functional [5,7]. Several efflux proteins from this family have been linked to MDR and ineffective cancer chemotherapy [5]. ABCB1 (also known as P-glycoprotein, P-gp, and MDR1), ABCG2 (also known as MXR, ABCP, and BCRP—breast cancer resistance protein), and ABCC1 (also known as MRP1—multidrug resistance protein 1) have been widely studied. Thirteen members of the ABCC family have been discovered, nine of which were characterized as multidrug resistance proteins (MRPs) including ABCC1 (MRP1), ABCC2 (MRP2), ABCC3 (MRP3), ABCC4 (MRP4), ABCC5 (MRP5), ABCC6 (MRP6), ABCC10 (MRP7), ABCC11 (MRP8), and ABCC12 (MRP9) [8]. Some antiepileptic drugs (AEDs) can act as substrates for P-gp or MRPs; therefore, the overexpression of such transporters at the blood–brain barrier (BBB) decreases the concentrations of these drugs in the brain, leading to refractory epilepsy with an MDR phenotype [9]. While P-gp extrudes xenobiotics, MRPs are responsible for the efflux of both endogenous agents and xenobiotics and are involved in physiological functions such as intracellular detoxification, oxidative stress, inflammation, and substance transport [8].

*ABCB5* is expressed in different tissues, including melanocytes, melanoma cells, testis, mammary tissue, and retinal pigmented epithelium. At the transcriptional level, *ABCB5* is expressed in malignant melanoma, breast cancer, colorectal cancer, hepatocellular cancer, and leukemia [5], and it is considered a marker of cancer stem cells, particularly in melanoma and colorectal cancer [10].

*ABCB5* encodes a full transporter (known as ABCB5FL, ABCB5.ts, expressed in the testis and prostate) and a half transporter (ABCB5β, expressed in melanocytes and melanoma). Transcript variants like ABCB5α are too short to be functional, and they would hypothetically have a regulatory role rather than a transport role [5,11]. Melanoma cells with high *ABCB5* expression show elevated metastatic potential in vitro and in vivo, and *ABCB5* mutations have been shown to promote proliferation and invasiveness capabilities in melanoma cells [12]. *ABCB5* may be expressed in melanoma stem cells with a CD133^+^ phenotype and might mediate DOX resistance [13,14]. In breast cancer, *ABCB5* is also overexpressed at the transcriptional level and can enhance metastasis and epithelial–mesenchymal transition (EMT) through the expression of ZEB1 [15].

The ABC transporter family’s efflux pumps are subject to epigenetic gene regulation [16]. HDACis have been shown to induce differentiation, arrest the cell cycle, and generate apoptosis in several transformed cell lines; they inhibit tumor growth in animal models and show antitumor activity in clinical trials [17]. However, HDACi treatment can activate the MDR phenotype by either naturally expressing it or inducing it via chemotherapy treatment [18]. Many clinical trials for improving the efficiency of chemotherapy through the inhibition of ABCB1 result in high levels of toxicity because it blocks the fundamental physiological roles of this protein [19]. VPA, an eight-carbon, branched-chain fatty acid, is a well-known antiepileptic drug and an effective HDACi that causes the hyperacetylation of the N-terminal H3 and H4 histone tails in vitro and in vivo [20,21]. Although VPA was classified as a class I (HDAC1, 2, 3, and 8) HDACi [22], it is still referred to as a pan-inhibitor [23].

In this work, we explored the intrinsic and 5 mM VPA-induced expression of ABC proteins related to drug resistance in HeLa cell cultures at the mRNA and protein levels. Furthermore, we determined if VPA-induced ABCB5 influences the acquisition of the MDR phenotype using DOX as a known substrate of the ABCB5 protein, which confers chemotherapy resistance in melanoma.

## 2. Materials and Methods

### 2.1. Chemical Reagents

VPA and 3-[4,5-dimethylthiazol-2-yl]-2,5-diphenyl tetrazolium bromide (MTT) were purchased from Sigma Chemical Co., (St. Louis, MO, USA), while DOX was purchased from Santa Cruz Biotechnology (Santa Cruz, CA, USA).

### 2.2. Cell Culture

The HeLa cell line from cervical cancer CCL-2 was kindly donated by Dr. Saúl Villa Treviño, a researcher at the CINVESTAV, and was cultured in Dulbecco’s modified Eagle’s medium (DMEM Glutamax, Gibco, Life-Technologies, Invitrogen, Grand Island, NE, USA) supplemented with 10% (*v*/*v*) heat-inactivated fetal bovine serum (FBS, Biowest, Kansas City, MO, USA) and an antibiotic–antimycotic solution (Gibco). The cell cultures were maintained at 37 °C in a humidified atmosphere containing 5% CO_2_.

### 2.3. Total RNA Extraction, Reverse Transcription, and Polymerase Chain Reaction to Evaluate the Expression of Genes Related to Multidrug Resistance Mechanisms

Total RNA was isolated with TRIzol (Invitrogen, Grand Island, NE, USA) according to the manufacturer’s instructions. RNA integrity was electrophoretically verified with 1% agarose gel by ethidium bromide staining, and RNA purity was obtained to be ˃1.8 using the OD_260_/OD_280_ nm absorption ratio. The total RNA (0.5 µg) of untreated HeLa cells or HeLa cells treated with 5 mM VPA, a concentration close to the IC_50_ of VPA (5.8 mM) [24], was reverse-transcribed using the Maxima First Strand cDNA Synthesis kit (Thermo Scientific, Vilnius, Lithuania), and a polymerase chain reaction (PCR) was carried out using PCR Master Mix (2X) (Thermo Scientific, Vilnius, Lithuania). The primer sequences used for this study are shown in Table 1, with GAPDH primers as the housekeeping gene control. The thermal cycling conditions for each primer set were as follows: 95 °C for 3 min; 35 cycles of the denaturation step at 95 °C for 30 s, annealing at 50 °C for 30 s, and extension at 72 °C for 30 s; and lastly, a final extension cycle at 72 °C for 5 min. PCR products were resolved on a 3% agarose gel using ethidium bromide staining.

### 2.4. ABCB5 Sequencing

To verify the identity of the ABCB5 fragment, the 85 bp band was cloned into competent *E. coli* DH5α cells using the CloneJET PCR Cloning Kit (Thermo Scientific, Vilnius, Lithuania) and sequencing. The sequence was analyzed by the BLAST + version 2.8, 2.11.0, 2.13.0 NCBI program and showed 100% homology to *ABCB5* variants 1 and 2.

### 2.5. ABCB5 Immunofluorescence Staining

HeLa cells (15 × 10^3^ cells) were cultured on coverslips for 24 h and treated for the subsequent 48 h with a fresh medium containing 5 mM VPA. The cells were fixed with 100% ice-cold methanol, blocked with 5% normal goat serum, and incubated overnight at 4 °C with a 1:200 dilution of primary ABCB5 monoclonal antibody (Novus Biologicals, Littleton, CO, USA). The immunogen refers to amino acids 481–674 of the ABCB5 isoform beta. Later, the cells were incubated with a 1:200 dilution of goat anti-mouse IgG-FITC (Santa Cruz, CA, USA) for 2 h at room temperature, protected from the dark, and counterstained with Vectashield sealant containing 4′,6-diamidino-2-phenylindole (DAPI). The samples were examined under a confocal microscope, Olympus Fluoview Fv300 (Nagano, Japan), and the pictures were scanned with space dimensions of X-Y-Z.

### 2.6. Trypan-Blue Exclusion Method

HeLa cells (5 × 10^3^ cells per well) in 500 µL of DMEM containing 10% FBS were seeded in 24-well culture plates. After 24 h, HeLa cells were treated with a fresh medium containing 5 mM VPA or DMSO (vehicle), and the plates were incubated for 24, 48, and 72 h. The number of living cells per well was determined by the Trypan-blue exclusion method as follows: adherent and floating cells were collected, and 0.4% Trypan-blue solution (Sigma-Aldrich, St. Louis, MO, USA) was added in a 1:1 (*v*/*v*) ratio and counted in a TC10 Automated Cell Counter (BIO-RAD, Laboratories, Hercules, CA, USA). Doubling time was calculated with 0, 24, and 48 h data [25].

### 2.7. In Vitro Cytotoxicity Assay of DOX for VPA Sensitization of HeLa Cells

HeLa cells (1 × 10^3^ cells/well) in 100 µL of DMEM supplemented with 10% FBS were seeded in 96-well culture plates. After 24 h, some wells were pretreated with a fresh medium containing 5 mM VPA for an additional 24 h. Subsequently, pretreated cells were cotreated with 5 mM VPA and different concentrations of DOX (0.25, 0.5, 1, and 1.5 µM) for 24 h, and other cells were also treated with DOX alone at the same concentrations. The plates were analyzed for cell survival using the colorimetric 3-(4,5-dimethylthiazol-2-yl)-2,5-diphenyltetrazolium bromide (MTT) dye reduction assay (Sigma-Aldrich, St. Louis, MO, USA), as described in another study [24].

### 2.8. Statistical Analysis

Data were expressed as the mean ± SD. A two-way ANOVA followed by Sidak’s multiple comparisons test was performed, using GraphPad Prism version 8.4.3 for Windows, GraphPad Software, La Jolla, CA, USA (www.graphpad.com) (accessed on 13 April 2025).

## 3. Results

### 3.1. The Epigenetic Effect of 5 mM Valproic Acid Changes the Expression of the ABCB1 and ABCB5 Pumps in HeLa Cells

To examine the expression of drug resistance pumps when treated with the epigenetic agent, cDNA from HeLa cells incubated with 5 mM VPA was amplified by PCR, with the results showing that these cells increase the expression of *ABCB1* and induce the de novo expression of *ABCB5*. HeLa cells intrinsically express a wide variety of ABC family resistance genes, including those encoding the Pgp, MRP1–7, and BCRP protein, but they do not express *ABCB5* (Figure 1). Treatment with 5 mM VPA increases the expression of *ABCB1* and induces the de novo expression of *ABCB5* in HeLa cells treated for 24 h.

### 3.2. Sequencing of the ABCB5 Fragment Obtained by RT-PCR Demonstrates That It Is an Amplicon of ABCB5 cDNA

To confirm that the 85 bp sequence amplified using primers for *ABCB5* was indeed of this sequence type, capillary sequencing was carried out, which showed 100% homology with the cDNA sequence to *ABCB5* variants 1 and 2 corresponding to ABCB5FL and ABCB5β, respectively (Figure 1).

### 3.3. It Is Possible to Detect the ABCB5 Pump Using a Specific Monoclonal Antibody in HeLa Cells Treated with 5 mM Valproic Acid

Given the importance of ABCB5 in human cancer biology, we explored the expression of this pump at the protein level in HeLa cells treated with close the IC_50_ of VPA. Using a specific monoclonal antibody against ABCB5, an immunocytochemistry assay was performed with this antibody, demonstrating that cells expressing ABCB5 mRNA also synthesize the pump protein. HeLa cells that receive no treatment do not exhibit ABCB5 protein expression. This protein was detected by immunocytochemistry in HeLa cells treated with 5 mM VPA for 48 h (Figure 2). This is the first report demonstrating that treatment with VPA induces *ABCB5* expression at both the mRNA and protein levels in HeLa cells.

### 3.4. The Presence of the ABCB5 Pump in HeLa Cells Is Not Sufficient to Confer an Advantage Against Treatment with DOX

In this work, despite the expression of *ABCB5* and the overexpression of *ABCB1*, treatment with 5 mM VPA reduces the growth rate by 20%, delays cell duplication time by 25% and decreases the number of living cells per well to approximately 50% after 72 h in HeLa cells (Figure 3). To assess whether the ABCB5 protein detected in HeLa cells due to 5 mM treatment could confer chemotherapy resistance, HeLa cells were pretreated with 5 mM VPA for 24 h, followed by 24 h of DOX cotreatment, which reduced the DOX IC_50_ value in half (0.25 µM vs. 0.5 µM alone) (Figure 4).

## 4. Discussion

Valproic acid has been shown to induce P-glycoprotein in human tumor cell lines and in rat liver [26], while ABC transporters have been implicated in chemoresistance and treatment failure in acute myeloid leukemia. In acute myeloid leukemia (AML), an increased expression of BCRP is associated with an increased risk of relapse and reduced survival, particularly when combined with Pgp [27]. Approximately 50% of AML patients have blast cells that express *ABCB1*, and its overexpression is associated with lower complete remission (CR) rates; treatment with chemotherapy combining an anthracycline with cytarabine has higher relapse rates. ABCB1 activity did not directly mediate chemotherapy resistance, and ABCB1 inhibition did not improve the outcome of AML patients with overexpressed *ABCB1*, but this result may be useful in clinical routine for improving the chemotherapy selection [28]. On the other hand, midostaurin, an ABCB1 inhibitor, increases anthracycline accumulation in peripheral blood mononuclear cells of CD34^+^ AML patients who do not achieve CR [29].

In melanoma, subpopulations of ABCB5-expressing cells increase their tumorigenicity and display properties similar to those of cancer stem cells (CSCs) [30], which contribute to cancer initiation, progression, metastasis, recurrence, and resistance to chemotherapy [20,21,31]. However, most previous studies have not specified the form investigated, and the data provided do not allow for discrimination between ABCB5FL and ABCB5β. ABCB5FL transports anthracyclines, taxanes, vinca-alkaloids, and epipodophyllotoxin. In contrast, no drugs have been identified as substrates for the β isoform, likely due to the limited number of molecules tested [11]. ABCB5β is predicted to contain a TMD composed of six α-helices flanked by two NBDs, with one being complete and the other being truncated. Motifs of potential dimerization have been identified in ABCB5β, suggesting that it might hetero-dimerize (with ABCB6 or ABCB9) to form a functional transporter [32].

The anticancer effect of VPA is primarily attributed to the inhibition of HDACs [14]. In HeLa cells, a 5 mM VPA treatment inhibits approximately 90% of the deacetylase activity [33]. This treatment also inhibits HeLa cell proliferation by arresting the cell cycle in the G0/G1 phase, which occurs as a result of the overexpression of *p21* without caspase-3 activation [33]. *ABCB1* overexpression and *ABCB5* expression may act as epigenetic mechanisms that increase the histone acetylation level at both genes. While DOX is a known ABCB5 substrate that induces MDR in melanoma cells [14], this effect was not observed in this study. In contrast, pretreatment with 5 mM VPA and cotreatment with DOX sensitized HeLa cells, reducing the IC_50_ of DOX to half when compared to DOX treatment only (Figure 4). Pretreatment with trichostatin A (TSA) or SAHA, two HDACis, can increase the cytotoxicity of anticancer drugs, even in intrinsically resistant cells [34]. In patients with primary tumors of breast cancer, hydralazine (a DNA demethylation agent) and magnesium valproate up- and downregulate 1091 and 89 genes, respectively, by at least 3-fold; *ABCB5* was only downregulated [35]. VPA has been found to sensitize gemcitabine-induced cytotoxicity in gemcitabine-resistant pancreatic cancer cells (PCCs) in vitro [36]. In another report, VPA induces tumor migration and invasion through EMT in colon cancer cell lines [37], which promotes the malignant progression of tumors. However, low-dose VPA (0.5 mM), in combination with gemcitabine, promotes the migration and invasion of pancreatic cancer, and high-dose VPA (5 mM) enhances the sensitivity of PCCs to gemcitabine through p38 activation, which suppresses the activation of STAT3 and Bmi1 [38].

Human ABC transporters are located in the plasma membrane and have transport functions [39]. As seen in Figure 2, the ABCB5 protein expressed due to the 5 mM VPA treatment suggests that it is not located in the plasma membrane, but it might be in the endoplasmic reticulum, since it was recently found that the β isoform induced by SAHA is located in the endoplasmic reticulum, and this isoform has no transport function [7,11]. The expressed isoform in our model possibly lacked pump activity and could not transport DOX out of the cell.

Based on the results of this work, we propose that HDACis, like VPA, can sensitize cancer cells, as the induced expression of certain ABC proteins does not confer functionality. Additionally, treatment with these epigenetic drugs may cause cells to express novel targets, such as ABCB5, which could be advantageous in biological therapies using therapeutic monoclonal antibodies.

## Figures and Tables

**Figure 1 cimb-47-00749-f001:**
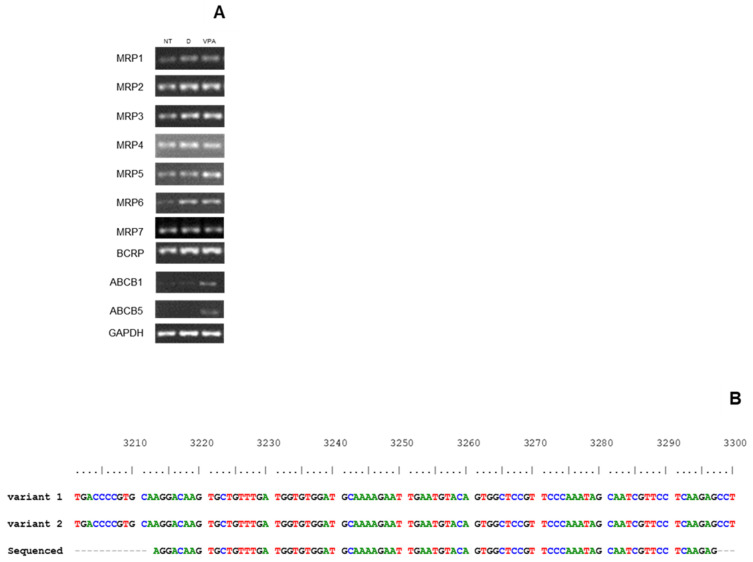
RT-PCR of some genes related to multidrug resistance mechanisms (**A**); the 85 bp band was cloned and sequenced (**B**). NT: non-treated; D: DMSO; VPA: valproic acid.

**Figure 2 cimb-47-00749-f002:**
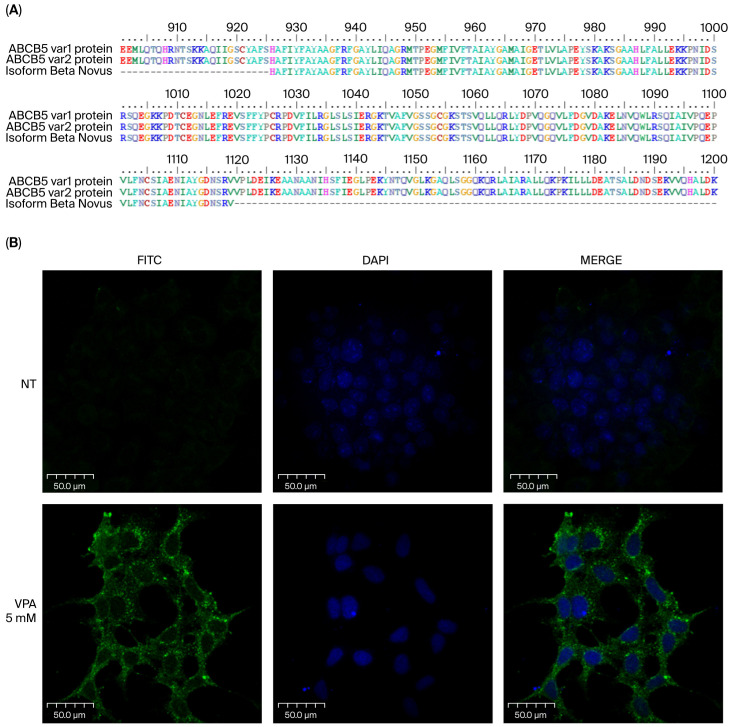
Protein sequence detected by Novus ABCB5 monoclonal antibody and its comparison to variants 1 and 2. The immunogen comprises 194 amino acids (**A**). ABCB5 protein immunodetection in HeLa cells treated with 5 mM VPA (**B**). NT: non-treated.

**Figure 3 cimb-47-00749-f003:**
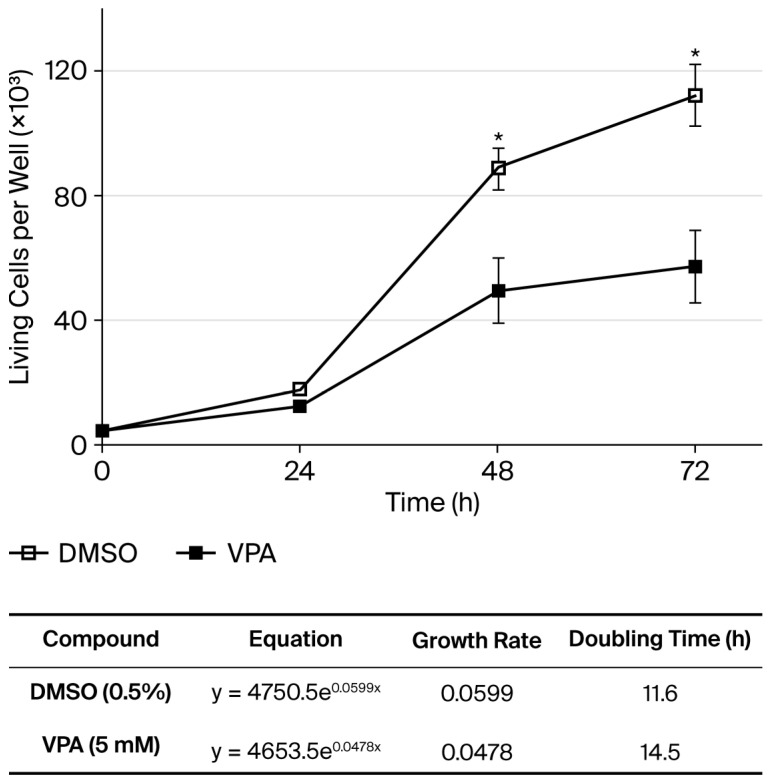
Living HeLa cells per well treated with 5 mM VPA for 24, 48, and 72 h, showing the equation (R^2^ = 0.9962 and 0.9884 for DMSO and VPA, respectively), growth rate, and doubling time. The living cells were determined by Trypan blue and are expressed as the mean ± SD for n = 3. * *p* value 0.0351 (48 h) and 0.0149 (72 h).

**Figure 4 cimb-47-00749-f004:**
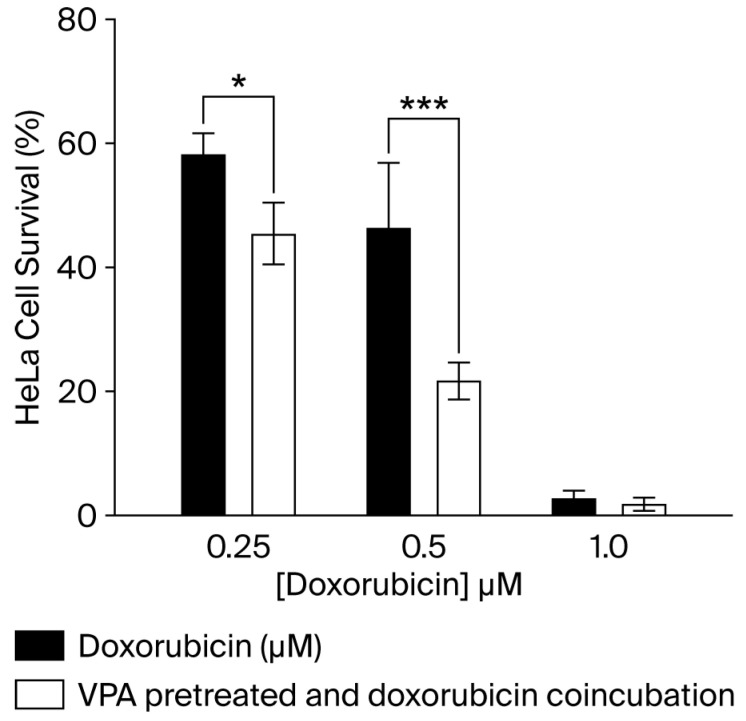
Effect of pretreatment and coincubation of 5 mM VPA with different concentrations of DOX on HeLa cell survival determined by MTT assay. Results are expressed as the mean ± SD for n = 3. * *p* value 0.0301; *** *p* value 0.0002.

**Table 1 cimb-47-00749-t001:** Oligonucleotides used to evaluate the expression of genes related to multidrug resistance mechanisms.

	Gene (Protein)	Forward	Reverse	Size (bp)
1	*ABCC1* (MRP1)	5′-gACAgAgATTggCgAgAAg-3′	5′-ATgTCAgCgTTggAgTACAC-3′	90
2	*ABCC2* (MRP2)	5′-AgAgAAgCTgACCATCATCC-3′	5′-ACCAggATCTTggATTTCCg-3′	255
3	*ABCC3* (MRP3)	5′-TgCAggTgACATTTgCTCTg-3′	5′-AgCgCACAgAATAATTCCgg-3′	192
4	*ABCC4* (MRP4)	5′-CTgAgAATgACgCACAgAAg-3′	5′-TATgggCTggATTACTTTgg-3′	122
5	*ABCC5* (MRP5)	5′-TCCgCCACTgTAAgATTCTg-3′	5′-CATggCATAgAATCgggAAC-3′	242
6	*ABCC6* (MRP6)	5′-TgCAgTACAAgTgTgCTgAC-3′	5′-CTgTAAAACAggCCCTTCTg-3′	313
7	*ABCC10* (MRP7)	5′-gAggTgATTACATCCATggg-3′	5′-ACTgTCTTgTTggCAAAgCg-3′	212
8	*ABCG2* (BCRP)	5′-AAggAgATCAgCTACACCAC-3′	5′-CACTgCTgAAACACTggTTg-3′	232
9	*ABCB1* (MDR1, Pgp)	5′-TACTTggTggCACATAAACTC-3′	5′-gCATAgTCAggAgCAAATgA-3′	113
10	*ABCB5*	5′-AggACAAgTgCTgTTTgATg-3′	5′-CTCTTgAggAACgATTgCTA-3′	85
11	*GAPDH*	5′-gTATgACAACAgCCTCAAgAT-3′	5′-gTCCTTCCACgATACCAAAg-3′	104

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
