# Peer review of "Valproic Acid as a Histone Deacetylase Inhibitor Induces ABCB1 Overexpression and De Novo ABCB5 Expression in HeLa Cells"

_cimb, 2025, doi:10.3390/cimb47090749_

Round 1

Reviewer 1 Report

Comments and Suggestions for Authors

CIMB – 3806192

August 03, 2025

This manuscript demonstrates that treatment with 5 mM sodium valproate (VPA) for 48 h induces the expression of ATP-binding cassette protein ABCB5 and increases the ABCB1 expression in HeLa cells, but that the presence of ABCB5 does not induce these cells to acquire a multidrug resistance phenotype. When using doxorubicin as a substrate for the ABCB5 protein, which confers chemotherapy resistance in melanoma, no advantage was demonstrated for HeLa cells.

This study is interesting but requires that discussion of results should be primarily centered mostly into effects on HeLa cells and the potential action of these results on cervical cancer, as authors have chosen HeLa cells for the demonstration of ABC proteins. Conversely, most of the Discussion text reports other tumor cell types making the rational confuse.

Why studies with VPA concentrations less than 5 mM were not carried out?

Abstract: Please write MDR in full the first time it is mentioned in the text or write (MDR) before “pumps”; write (ABC) after “cassette”

Introduction: line 41: substitute “but” for “and”

lines 78-80: write HDACi, HDACI in full the first time these terms are mentioned

line 86 – VPA is not only an HDAC inhibitor but also an agent that affects the DNA methylation status especially in HeLa cells

Materials and Methods: line 104 – Its is mentioned that HeLa cell lines were used. Which cell lines?

line 117 -Thermoscientific: inform city and country

  1. 3 – Standardize Thermoscientific or Thermo Scientific

Discussion: Only at line 249 an abbreviation of doxorubicin was used. You may choose to use this abbreviation early, at the Introduction section, and then mantain it alongside the text.

References: Please revise citation format

English revision is required.

Comments on the Quality of English Language

English revisison is required.

Author Response

ANSWERS TO REVIEWER 1

This manuscript demonstrates that treatment with 5 mM sodium valproate (VPA) for 48 h induces the expression of ATP-binding cassette protein ABCB5 and increases the ABCB1 expression in HeLa cells, but that the presence of ABCB5 does not induce these cells to acquire a multidrug resistance phenotype. When using doxorubicin as a substrate for the ABCB5 protein, which confers chemotherapy resistance in melanoma, no advantage was demonstrated for HeLa cells.

1.This study is interesting but requires that discussion of results should be primarily centered mostly into effects on HeLa cells and the potential action of these results on cervical cancer, as authors have chosen HeLa cells for the demonstration of ABC proteins. Conversely, most of the Discussion text reports other tumor cell types making the rational confuse.

Cancer chemoresistance due to ABC proteins is one of the primary obstacles to successful anticancer treatments. Furthermore, it has been widely reported that the use of HDACi induces the expression of these proteins. Our initial objective was to explore innate and acquired resistance using the HDACi treatment, specifically with VPA, in HeLa cells as a representative cellular model of epithelial cancer, since it constitutes 90% of diagnostics in patients. In this work, we unexpectedly found de novo expression of ABCB5 in HeLa cells due to 5 mM VPA treatment. There are no reports of the expression of ABCB5 in HeLa cells, so in the Discussion, it was necessary to consider the case of melanoma, where ABCB5 was found and characterized. ABCB5 regulation and function are not yet fully understood, so this finding could be a good model for its study.

  1. Why studies with VPA concentrations less than 5 mM were not carried out?

In previous work, we found that the IC50 for VPA in HeLa cells with MTT was 5.8 mM, so we decided to use 5 mM VPA, a concentration close to the IC50 obtained. The reference is cited in the short report that you are kindly reviewing, on line 113 corresponding to reference 24, and is shown below:

  1. Luna-Palencia, R.; Martinez-Ramos, F.; Vásquez-Moctezuma, I.; Fragoso-Vazquez, M.J.; Mendieta-Wejebe, J.E.; Padilla-Martínez, I.I.; Sixto-Lopez, Y.; Mendez-Luna, D.; Trujillo-Ferrara, J.; Meraz-Rios, M.A.; Fonseca-Sabater, Y.; Correa-Basurto, J. Three amino acid derivatives of valproic acid: design, synthesis, theoretical and experimental evaluation as anticancer agents. Anticancer Agents Med Chem 2014;14: 984-993.

Reference 24 is available at the following link:

https://drive.google.com/drive/folders/13MS0vbl4Q6scQrhuPpp6oiT5aZi9W6_n?usp=sharing

It might be interesting to explore the ABCB5 expression with different concentrations of VPA but since this is a brief report, our objective is to report that the treatment of HeLa cells with 5 mM VPA induces the expression of this protein, related to multidrug resistance in melanoma but not in HeLa cells. This finding could provide a model for the study of its regulation and function.

  1. Abstract: Please write MDR in full the first time it is mentioned in the text or write (MDR) before “pumps”; write (ABC) after “cassette”

Thanks, we have corrected this.

  1. Introduction: line 41: substitute “but” for “and”

Thanks, but we could not find “but” in line 41.

  1. lines 78-80: write HDACi, HDACI in full the first time these terms are mentioned

Thanks, we have corrected this.

  1. line 86 – VPA is not only an HDAC inhibitor but also an agent that affects the DNA methylation status especially in HeLa cells

That is right; since 2007 it was found that VPA causes changes in the state of DNA methylation of genomic DNA: doi: 10.1093/carcin/bgl167. However, most gene expression studies in cancer cells have been conducted considering VPA as a histone deacetylase inhibitor (HDACi). Likewise, in our previous work (Reference 33: Figure 4), we found that with 5 mM VPA, the HDAC activity was reduced by 90% in HeLa cells compared to a control without VPA treatment, so that VPA showed an HDACi effect at 5 mM.

Reference 33 is available at the following link:

 https://drive.google.com/drive/folders/13MS0vbl4Q6scQrhuPpp6oiT5aZi9W6_n?usp=sharing)

  1. Materials and Methods: line 104 – Its is mentioned that HeLa cell lines were used. Which cell lines?

The human cervical cancer cell line HeLa corresponds to ATCC CCL-2.  

  1. line 117 -Thermoscientific: inform city and country
  1. 3 – Standardize Thermoscientific or Thermo Scientific

Thanks, we have corrected this (Thermo Scientific, Lithuania) on lines 114, 115, and 123.

  1. Discussion: Only at line 249 an abbreviation of doxorubicin was used. You may choose to use this abbreviation early, at the Introduction section, and then mantain it alongside the text.

Thanks, doxorubicin (DOX) is now found in the Abstract, line 21, and we have substituted all uses of “Doxorubicin” for “DOX” throughout the text. 

  1. References: Please revise citation format

Thanks, we have reviewed the citation format.

Reviewer 2 Report

Comments and Suggestions for Authors

This report provides evidence that valproic acid (VPA), an epigenetic modulator inhibiting class I histone deacetylases, activates de novo expression of ABCB5 transporter gene (in Hela cells cultures) and thus may contribute to cancer cell resistance to chemotherapy. While the VPA role as ubiquitous activator of gene transcription due to chromatin relaxation is well established, direct link between the HDAC1 inhibitor and ABCB5 expression has not been well established. In this respect, the short scientific report has a merit for publication.

 However, there are numerous shortcomings that need to be overcome by the authors in order to generate a respectable scientific report.

1). Hela cell line is a good model system to demonstrate the effect of VPA as a stimulator of abcb5 expression given that the base line of abcb5 gene transcription in this cell type is undetectable.

2). However, HeLa cells are not the right model to investigate the effect of VPA as inducer of MDR since this HDACi causes cell cycle arrest and apoptosis in this cell type [DOI: 10.3892/or.2013.2747 ]. Thus, the primary goal of the study aiming to demonstrate the controversial role of VPA as supplementary to the anticancer chemotherapy treatment cannot be achieved using Hela cell. Even clinically more relevant to the physiological role of ABCB5 cell lines, including melanoma (A-375 or SK-MEL3) and colorectal cancer (HCT116 or Caco-2), cannot clearly demonstrate the MDR effect of VPA since the HDACi induces apoptosis in all above-mentioned cancer cell lines. Therefore, the complimentary effect of VPA to doxorubicin is expected and has been partially demonstrated in this study (Fig.4).

3). Fig 3 in this study demonstrates the effect of VPA on HeLa cell survival and not the division rates as stated. Trypan blue enters cells with compromised membranes. Cell division can occur at significantly earlier time points. Therefore, all references to division rate (including table under the figure) are incorrect. If the authors aspire to show the effect of VPA on cell division rate other well-established methods should be utilized (cell dilution of CFSE, DNA synthesis analysis with BrdU, or Ki67 expression via immunocytochemistry).

4). If you wish to retain the discussion on the VPA role as inducer of MDR, more experiments need to be performed and presented, such as the use of fluorescently labeled DOX and quantitative analysis of VPA treated cells utilizing flow cytometry 6-24h post VPA treatment. If you cannot perform these experiments, remove any reference to VPA as a potential MDR inducer.

Comments on the Quality of English Language

The English grammar is very poor here and makes it difficult to comprehend the statements made by the authors. Please, improve English grammar and sentence structure using on-line editing programs in order to present a respectable scientific study.

Author Response

ANSWERS TO REVIEWER 2

This report provides evidence that valproic acid (VPA), an epigenetic modulator inhibiting class I histone deacetylases, activates de novo expression of ABCB5 transporter gene (in Hela cells cultures) and thus may contribute to cancer cell resistance to chemotherapy. While the VPA role as ubiquitous activator of gene transcription due to chromatin relaxation is well established, direct link between the HDAC1 inhibitor and ABCB5 expression has not been well established. In this respect, the short scientific report has a merit for publication.

 However, there are numerous shortcomings that need to be overcome by the authors in order to generate a respectable scientific report.

1). Hela cell line is a good model system to demonstrate the effect of VPA as a stimulator of abcb5 expression given that the base line of abcb5 gene transcription in this cell type is undetectable.

Our initial objective was to explore innate and acquired resistance by the HDACi treatment, specifically with VPA, in HeLa cells as a representative cellular model of epithelial cancer, since carcinoma constitutes 90% of patient diagnoses. In this work, we found unexpectedly de novo expression of ABCB5 in HeLa cells due to 5 mM VPA treatment, since as you mentioned in comment 2, this expression has been found in melanoma and colorectal cancer but not in HeLa cells and it has been related to MDR. Because ABCB5 is not detectable in HeLa cells neither at the mRNA nor protein level, we consider relevant that the use of 5 mM VPA can generate a study model of the regulation of ABCB5 expression, as well as to understand its function as a pump to extrude drugs, which contributes to resistance in cancer therapy.

2). However, HeLa cells are not the right model to investigate the effect of VPA as inducer of MDR since this HDACi causes cell cycle arrest and apoptosis in this cell type [DOI: 10.3892/or.2013.2747 ]. Thus, the primary goal of the study aiming to demonstrate the controversial role of VPA as supplementary to the anticancer chemotherapy treatment cannot be achieved using Hela cell. Even clinically more relevant to the physiological role of ABCB5 cell lines, including melanoma (A-375 or SK-MEL3) and colorectal cancer (HCT116 or Caco-2), cannot clearly demonstrate the MDR effect of VPA since the HDACi induces apoptosis in all above-mentioned cancer cell lines. Therefore, the complimentary effect of VPA to doxorubicin is expected and has been partially demonstrated in this study (Fig.4).

In the discussion of the article, you cited [DOI: 10.3892/or.2013.2747], it is mentioned that: “VPA decreased the growth of HeLa cells in dose- and time-dependent manners. When the cell cycle distributions were examined, 10 mM VPA induced a G2/M phase arrest of the cell cycle in HeLa cells at 24 h”. But they also mentioned: “However, relatively lower concentrations of VPA seemed to induce a G1 phase arrest in HeLa cells”. In our work, 5 mM VPA was uses since, in a previous study, which we cite in this short report that you are kindly reviewing as reference 33, we found that at this concentration in HeLa cells, p21 expression increases  (Figure 6A), inducing cell cycle arrest in the G0/G1 phase and showing a decrease in apoptosis(Figure 6B); without increased caspase-3 activity (Figure 6C).

Reference 33 available at the following link:

 https://drive.google.com/drive/folders/13MS0vbl4Q6scQrhuPpp6oiT5aZi9W6_n?usp=sharing)

  1. Luna-Palencia GR, Correa-Basurto J, Trujillo-Ferrara J, Meraz-Ríos MA, Vásquez- Moctezuma I. Epigenetic Evaluation of N-(2-hydroxyphenyl)-2-Propylpentanamide, a Valproic Acid Aryl Derivative with Activity Against HeLa Cells. Curr Mol Pharmacol. 2021 Oct 25;14(4):570-578. doi: 10.2174/1874467213666200730113828. PMID: 32744980.

Therefore, we consider HeLa cells as good model because treatment with 5 mM VPA does not induce apoptosis. As you mentioned, further experiments are needed to determine the functionality of the ABCB5 protein as a drug extrusion pump but in that case, the objective of this short report is communicating the ATP-binding cassette protein expression profile in untreated and treated-VPA HeLa cells. Furthermore, as mentioned in this short report based on reference 39, human ABC transporters with transport function, are in the plasma membrane, and Figure 2 shows the possibility that the VPA-induced ABCB5 isoform may be in the endoplasmic reticulum. This could explain why this isoform, located in the endoplasmic reticulum, is not functional for transport, as recently reported by Divivier L et al., 2022 and Díaz-Anaya AM et al., 2023; these authors are cited in this short report as references 7 and 11, respectively.

On the other hand, Figure 4 shows the effect of pretreatment with VPA and subsequent cotreatment with doxorubicin, which results in the sensitization of HeLa cells to DOX, obtaining a 50% reduction in the DOX IC50, which could impact chemotherapy treatments.

3). Fig 3 in this study demonstrates the effect of VPA on HeLa cell survival and not the division rates as stated. Trypan blue enters cells with compromised membranes. Cell division can occur at significantly earlier time points. Therefore, all references to division rate (including table under the figure) are incorrect. If the authors aspire to show the effect of VPA on cell division rate other well-established methods should be utilized (cell dilution of CFSE, DNA synthesis analysis with BrdU, or Ki67 expression via immunocytochemistry). 

The objective of this experiment was to determine the effect of 5 mM VPA on the number of live HeLa cells compared to untreated cells. Trypan blue is an exclusion dye that penetrates dead cells whose cell membrane is not intact and does not penetrate live cells with an intact cell membrane. With this method, indirectly, that is, by quantifying total and dead cells, we can quantitatively know the number of living cells since the cell counter used shows total cells, living cells, and percentage viability. An image of the cell counter screen is shown below.

PLEASE SEE THE PDF

With the number of living cells obtained and considering the sample volume, a growth curve was constructed with the live cells per well vs time, and the reduction in the number of live cells per well due to treatment with 5 mM VPA, is clearly observed. With the software used, cited as reference 25: Roth V. 2006 Doubling Time Computing, Available from: http://www.doubling-time.com, the data for times 0, 24 and 48 h fit perfectly to an exponential model (R2 = 0.9962 and 0.9884 for DMSO and VPA, respectively). With the corresponding equation, for example for DMSO, y = 4750.5 e 0.0599x, the growth rate was obtained as the exponent of e, that is, 0.0599 and the doubling time is ln 2 / growth rate, that is, ln 2 / 0.0599. We showed that due to 5 mM VPA, the time for a HeLa cell to duplicate is increased respect to the untreated cell.

4). If you wish to retain the discussion on the VPA role as inducer of MDR, more experiments need to be performed and presented, such as the use of fluorescently labeled DOX and quantitative analysis of VPA treated cells utilizing flow cytometry 6-24h post VPA treatment. If you cannot perform these experiments, remove any reference to VPA as a potential MDR inducer.

The role of VPA as MDR inducer was demonstrated in several reports (doi:10.3892/ol.2014.2714;  More experiments are needed to study the functionality of the 5 mM VPA-induced ABCB5 isoform in HeLa cells but the objective of this short report is communicate that work as a model for study its regulation and its functionality. The novelty was the unexpected expression of ABCB5 without the MDR phenotype. (doi: 10.3892/ol.2014.2714)

Round 2

Reviewer 1 Report

Comments and Suggestions for Authors

Improvement of the manuscript was sufficient.

Introduction: In English, substitute "but" for "and" means change "and" into "but"

Author Response

Introduction: In English, substitute "but" for "and" means change "and" into "but"

Before English Editing Service of MDPI:

ABC transporters not only are expressed by cancer cells and also are located in cell membrane of many normal tissues for protecting the cell from cytotoxic agents[3, 4].

After English Editing Service of MDPI:

ABC transporters are not only expressed by cancer cells; they are also located in the cell membrane of many normal tissues to protect the cell from cytotoxic agents [3,4].

Reviewer 2 Report

Comments and Suggestions for Authors

The revised version of this short scientific report is better constructed with improved English gramma and clear statements of the results. However, there is one statement made straight in the Abstract that is not supported by the results presented by this study and is most probably wrong: “Therefore, it is possible that the ABCB5 isoform induced by VPA treatment lacks MDR activity and does not confer the resistance phenotype”.

Although VA as HDACi is known to induce MDR through activation of ABC-transporter gene expression in certain cancer types such as HeLa, melanomas and colorectal cancers, VA has cytotoxic effect that overrides the MDR.

What this study shows is a synergistic effect between VA and DOX because both drugs exhibit cytotoxic effect on HeLa cells. When cells are dying there is no time to examine the MDR effect of the ABCB5 transporters.

Since this study does not provide proper experimental data on the ABCB5 activity, for example decreased amount of DOX in the cells, the statement in the Abstract regarding ABCB5 lacking MDR activity is premature and most probably wrong.

Minor revisions of the manuscript are necessary to make it suitable for publication, specifically removing the statement that VA induces ABCB5 isoform without MDR activity and adding the observation that VA acts synergistically with DOX due to their cytotoxic effects in this specific type of cancer cells.

Author Response

ANSWERS TO REVIEWER 2 ROUND 2, PDF WITH IMAGES

  1. The revised version of this short scientific report is better constructed with improved English gramma and clear statements of the results. However, there is one statement made straight in the Abstract that is not supported by the results presented by this study and is most probably wrong: “Therefore, it is possible that the ABCB5 isoform induced by VPA treatment lacks MDR activity and does not confer the resistance phenotype”.

We changed “Therefore, it is possible that the ABCB5 isoform induced by VPA treatment lacks MDR activity and does not confer the resistance phenotype” to “Probably the ABCB5 isoform induced by VPA treatment could lack MDR activity”.

  1. Although VA as HDACi is known to induce MDR through activation of ABC-transporter gene expression in certain cancer types such as HeLa, melanomas and colorectal cancers, VA has cytotoxic effect that overrides the MDR.

A substance or process that causes cell damage or death is referred to as cytotoxic. Cells exposed to cytotoxic compounds may undergo necrosis (uncontrolled cell death), apoptosis (programmed cell death), or autophagy, or stop actively growing and dividing to decrease cell proliferation (https://www.thermofisher.com/mx/es/home/life-science/cell-analysis/cell-viability-and-regulation/cytotoxicity.html#:~:text=Cytotoxicity%20is%20the%20degree%20to,in%20the%20cell%20culture%20medium).

For explaining the 5 mM VPA effect in HeLa cells, we present Figures 2B and 6B from the reference 33 (doi: 10.2174/1874467213666200730113828) available at the following link:

 https://drive.google.com/drive/folders/13MS0vbl4Q6scQrhuPpp6oiT5aZi9W6_n?usp=sharing)

Fig. (2). Effects of VPA and o-OH-VPA on the survival (A) and viability (B) of HeLa cells. Cells were incubated for 48 h at different concentrations of the compounds; the cell survival and viability were analyzed by MTT assay and the Trypan-blue exclusion method, respectively. Data are expressed as the mean ± SD for n ≥ 6 wells from at least two independent experiments for cell survival and n = 4 for viability.

Treatment of HeLa cells with 5 mM VPA for 24, 48, and 72 hours did not decrease cell viability, so the cytotoxic effect known as necrosis does not occur at this concentration (Fig. 2B). Furthermore, cell cycle analysis by flow cytometry shows that apoptosis is reduced in cells treated with 5 Mm VPA for 48 hours (Fig. 6B). The results shown in this report indicate that the observed reduction in living cell number following VPA exposure is attributable to a decreased growth rate and the increased duplication time.

We found that ABCB5 expression was induced at the mRNA level following 24 hours of 5 mM VPA treatment, with corresponding protein expression detected by immunocytochemistry at 48 hours. Crucially, cell viability remained at 100% at these critical time points (Fig. 2B of this report).

Considering that the effect of treating HeLa cells with 5 mM VPA is neither necrosis nor apoptosis, we have presented a model in which it is possible to study the functionality of the VPA-induced ABCB5 transporter. This model enables further investigation to determine whether this induction overcomes the MDR phenotype or if the resulting isoform is functionally incapable of drug efflux.

  1. What this study shows is a synergistic effect between VA and DOX because both drugs exhibit cytotoxic effect on HeLa cells. When cells are dying there is no time to examine the MDR effect of the ABCB5 transporters.

Synergy is a measure of drug interaction while sensitivity is a measure of drug combination efficacy. Sensitivity of a drug combination is defined as the level of treatment response, usually measured in the unit of percentage inhibition of cell viability or growth. In contrast, the synergy of a drug combination refers to the degree of drug interactions that contributes to the sensitivity of the drug combination independent of the single drug effects. Drug combination sensitivity may lead to a biased prioritization of drug combination that is unable to kill cancer cells despite strong synergy. (DOI:10.1371/journal.pcbi.1006752).

In this Short Report, we show the sensitization of HeLa cells to DOX by 5 mM VPA treatment, and not a synergistic effect between VPA and DOX. There are reports of sensitization of cancer cells by VPA (doi: 10.1016/j.dnarep.2017.08.002; https://doi.org/10.1038/s41598-017-15165-3)

We found that ABCB5 expression was induced at the mRNA level by 24 hours of 5 mM VPA treatment, with corresponding protein expression detected by immunocytochemistry at 48 hours. Crucially, cell viability remained at 100% at these critical time points (Fig. 2B of this Short Report).

  1. Since this study does not provide proper experimental data on the ABCB5 activity, for example decreased amount of DOX in the cells, the statement in the Abstract regarding ABCB5 lacking MDR activity is premature and most probably wrong.

In the Abstract, we changed “Therefore, it is possible that the ABCB5 isoform induced by VPA treatment lacks MDR activity and does not confer the resistance phenotype” to “Probably the ABCB5 isoform induced by VPA treatment could lack MDR activity”.

  1. Minor revisions of the manuscript are necessary to make it suitable for publication,

5a. specifically removing the statement that VA induces ABCB5 isoform without MDR activity and

It is likely that cells treated with 5 mM VPA express the β isoform. Our sequencing results support this, showing that the amplified fragment has 100% identity with the sequences of both the full-length (FL) and β isoforms. The FL isoform possesses transport activity, whereas the β isoform requires dimerization. Recently, the β isoform was reported to be localized to the endoplasmic reticulum (https://doi.org/10.3390/ijms242115847). The immunocytochemistry data in Figure 2 of this Short Report suggests that the ABCB5 protein is not localized to the plasma membrane.

This finding could provide a model for studying its regulation and function. Therefore, future studies may include colocalization experiments to determine whether it is an endoplasmic reticulum protein.

5b. adding the observation that VA acts synergistically with DOX due to their cytotoxic effects in this specific type of cancer cells

In this Short Report, we show the sensitization of HeLa cells to DOX by 5 mM VPA treatment, and not by a synergistic effect between VPA and DOX. There are reports of sensitization of cancer cells by VPA (doi: 10.1016/j.dnarep.2017.08.002; https://doi.org/10.1038/s41598-017-15165-3).
